# *Cryptococcus neoformans* Causing Meningoencephalitis in Adults and a Child from Lima, Peru: Genotypic Diversity and Antifungal Susceptibility

**DOI:** 10.3390/jof8121306

**Published:** 2022-12-16

**Authors:** Carolina Firacative, Natalia Zuluaga-Puerto, José Guevara

**Affiliations:** 1Studies in Translational Microbiology and Emerging Diseases (MICROS) Research Group, School of Medicine and Health Sciences, Universidad de Rosario, Bogota 111221, Colombia; 2Faculty of Natural Sciences, Universidad de Rosario, Bogota 111221, Colombia; 3Facultad de Medicina “San Fernando”, Universidad Nacional Mayor de San Marcos, Lima 15081, Peru

**Keywords:** antifungal susceptibility, cryptococcosis, *Cryptococcus neoformans*, MLST, Peru

## Abstract

Cryptococcosis, caused predominantly by *Cryptococcus neoformans*, is a potentially fatal, opportunistic infection that commonly affects the central nervous system of immunocompromised patients. Globally, this mycosis is responsible for almost 20% of AIDS-related deaths, and in countries like Peru, its incidence remains high, mostly due to the annual increase in new cases of HIV infection. This study aimed to establish the genotypic diversity and antifungal susceptibility of *C. neoformans* isolates causing meningoencephalitis in 25 adults and a 9-year-old girl with HIV and other risk factors from Lima, Peru. To identify the genotype of the isolates, multilocus sequence typing was applied, and to establish the susceptibility of the isolates to six antifungals, a YeastOne^®^ broth microdilution was used. From the isolates, 19 were identified as molecular type VNI, and seven as VNII, grouped in eight and three sequence types, respectively, which shows that the studied population was highly diverse. Most isolates were susceptible to all antifungals tested. However, VNI isolates were less susceptible to fluconazole, itraconazole and voriconazole than VNII isolates (*p* < 0.05). This study contributes data on the molecular epidemiology and the antifungal susceptibility profile of the most common etiological agent of cryptococcosis, highlighting a pediatric case, something which is rare among cryptococcal infection.

## 1. Introduction

Cryptococcosis is a life-threatening fungal infection of global distribution, caused by the encapsulated basidiomycetous yeasts *Cryptococcus neoformans* and *Cryptococcus gattii* species complexes [1]. These are ubiquitous microorganisms found in the environment, mainly in association with soil and pigeon guano, but also with decaying wood and plant material from various species of trees [2]. In the vast majority of cases of cryptococcosis worldwide, infection by the human immunodeficiency virus (HIV) is the main risk factor. Therefore, since the emergence of the HIV epidemic and the syndrome of acquired immunodeficiency (AIDS), the number of patients with cryptococcal infection in the world has increased dramatically [3]. In 2020, an estimated 152,000 people developed cryptococcal meningitis globally, resulting in 112,000 deaths, which accounted for 19% of AIDS-related mortality [4]. In addition, cryptococcosis continues to be an important mycosis in patients with other underlying conditions such as glucocorticosteroid treatment, solid-organ transplantation, hematologic malignancy, or other disorders associated with cell-mediated immune dysfunction [5].

In Peru, in 2021, around 98,000 people were living with HIV, with an estimated 84% of these people receiving antiretroviral therapy (ART) [6]. However, despite widespread availability of ART, there are about 5500 new cases of HIV infection per year in the country, and access to ART and health care services are still limited in many regions. As such, in Peru, cryptococcosis remains an important opportunistic infection, with high incidence and significant morbidity and mortality rates [7]. Furthermore, as it occurs in many other countries, in Peru, infections caused by *Cryptococcus* spp. are not of compulsory notification and the country lacks surveillance programs for invasive mycoses. Thus, precise and reliable information on local prevalence of cryptococcosis is not available. 

Since multilocus sequence typing (MLST) is a highly reproducible technique, used throughout the world for the molecular characterization of *C. neoformans* and *C. gattii* species complexes and for comparative studies between clinical, environmental and veterinary strains [8], it is possible to use this molecular methodology to characterize Peruvian isolates and to correlate the MLST genotypes with others reported in the country, Latin America and the world. Until now, only few *C. neoformans* isolates from Peru have been characterized molecularly. In one study, 32 isolates were genotyped by amplified fragment length polymorphisms (AFLP) fingerprinting, although no correlation of the Peruvian isolates with others from other countries was established, as this technique only determines the molecular type of the isolates [9]. Another study assessed, by MLST, the genetic diversity of 48 isolates recovered from 20 patients. Even though this study established the relationship between Peruvian isolates and those others recovered elsewhere, its aim rather focused on the investigation of isolates recovered from persistent cryptococcal infection to determine if patients were infected with the same genotype through time [10].

Despite the currently available options for the treatment of cryptococcosis, resistance to some antifungals has been described in isolates of both *C. neoformans* and *C. gattii* species complexes worldwide [11,12,13]. Therefore, in vitro susceptibility testing is useful, as it allows us to identify and monitor the emergence of reduced antifungal susceptibility, which in turn could provide us with relevant clinical information since resistance might hinder a successful therapy [14]. To contribute with data on the molecular epidemiology and the antifungal susceptibility profile of the most common etiological agent of cryptococcosis, the current study sets out to determine the MLST genotype and the antifungal susceptibility, to amphotericin B, four azoles and 5-fluorocytosine of *C. neoformans* isolates, recovered from 25 adults and a child, most of them with HIV, who were diagnosed with meningoencephalitis in three health care centers in Lima, Peru. These data will provide us with a more complete picture of the circulation and susceptibility profile of cryptococcal genotypes in Peru, which is rather limited compared to other countries in the region [15].

## 2. Materials and Methods

### 2.1. Isolates

Twenty-six *C. neoformans* isolates belonging to the fungal culture collection of the Instituto de Medicina Tropical Daniel A. Carrión from Universidad Nacional Mayor de San Marcos, Lima, Peru, were included in this study. The isolates were recovered between 2012 and 2021 from 26 patients who attended three health care institutions in Lima. From the patients, 19 (73.1%) were male and two (7.7%) were female (proportion 9.5:1), with ages ranging from 9 to 71 years (average 42.7 years). From five (19.2%) patients, data on sex and age were not available. All patients had a diagnosis of cryptococcal meningoencephalitis, which was made by cerebrospinal fluid (CSF) culture and by Indian ink staining of the CSF. The *Cryptococcus* species, recovered from a single colony, was identified by glycine assimilation on an L-canavanine-glycine-bromothymol blue medium, as described previously [16]. From the patients, 24 (92.3%) were HIV positive, including the pediatric case (a 9-year-old girl), and one each (3.8%) had diagnosis of cancer and a rheumatologic disease. Table 1 shows the list of isolates included in this study, the year of isolation and the age and sex of the patients, as well as data on genotyping (refer to Results section). Data on antifungal treatment, ART coverage and the outcomes of patients were not registered.

### 2.2. Multilocus Sequence Typing

After culturing each isolate on Sabouraud dextrose agar at 35 °C for 48 h, genomic DNA was extracted as described before [17]. Thereafter, the DNA amplification of six housekeeping genes, *CAP59*, *GPD1*, *LAC1*, *PLB1*, *SOD1*, *URA5* and the intergenic spacer region IGS1 was performed. This was done according to the International Society for Human and Animal Mycology (ISHAM) consensus MLST scheme for *C. neoformans* and *C. gattii* [8]. PCR products were commercially purified and sequenced, both forward and reverse strands, by Macrogen Inc., Seoul, Korea. Sequences were edited and contigs were assembled using the program Sequencher 5.4.6 (Gene Codes Corporation, Ann Arbor, MI, USA). Allele types (ATs) and sequence types (STs) were assigned to all isolates using the ISHAM consensus database, available at http://mlst.mycologylab.org/ (accessed on 10 November 2022). For the genotypic analysis, publicly available MLST data from 69 additional isolates from Argentina [18], Brazil [19,20,21], Colombia [22,23] and Peru [10] were included in order to compare our findings to those found in other countries in Latin America. These isolates were included as they represent all *C. neoformans* STs that have been reported so far in the region [15]. Concatenated sequences of the seven MLST loci of the 26 isolates from this study, together with the 69 Latin American isolates, were aligned with the software Mega 11. Using the same program, a dendrogram, showing the genetic relationship between the isolates, was constructed using maximum likelihood analysis with a bootstrap analysis of 100 replicates and the Jukes–Cantor model [24]. The establishment of the STs and the construction of the dendrogram allowed us as well to determine the molecular type (VNI and VNII) of the Peruvian isolates.

To estimate the genetic diversity of the *C. neoformans* clinical isolates from Peru, and based on the MLST results, the Simpson’s diversity index (D) was calculated with the number and frequency of STs found, not only for the whole population, but per molecular type. D values range from 0 to 1, where high scores (close to 1 or 1) indicate high diversity and low scores (close to 0 or 0) indicate low diversity [25]. The nucleotide diversity, Pi, and the haplotype (gene) diversity, Hd, of the population, and per molecular type, was also calculated, in pairwise comparisons, with the software DnaSP 6.12.03 [26].

The sequences obtained per isolate and per locus were deposited in GenBank under the following accession numbers: *CAP59* (OP856956–OP856981), *GPD1* (OP856982–OP857007), IGS1 (OP857008–OP857033), *LAC1* (OP857034–OP857059), PLB1 (OP857060–OP857085), *SOD1* (OP857086–OP857111), and *URA5* (OP857112–OP857137).

### 2.3. Antifungal Susceptibility Testing

Susceptibility of the isolates to six antifungal drugs was carried out using the Sensititre^®^ YeastOne^®^ plates (Thermo Scientific, Waltham, MA, USA), which is a colorimetric broth microdilution test, following the manufacturer’s instructions. Briefly, each isolate was grown on Sabouraud dextrose agar and incubated at 35 °C for 48 h. Subsequently, an inoculum, adjusted to a density of 0.5 McFarland standard (1–5 × 10^6^ cells/mL) was prepared, per isolate, in 5 mL of sterile water. From each yeast suspension, 20 µL were transferred into 11 mL of YeastOne^®^ inoculum broth to obtain a final concentration of 1.8–9 × 10^3^ cells/mL. From this last suspension, 100 µL were deposited in each well of the Sensititre^®^ YeastOne^®^ plate. Plates were sealed, incubated at 35 °C and read after 72 h of incubation. Colorimetric minimum inhibitory concentration (MIC) was defined as the lowest concentration of each antifungal that prevented the development of a pink or fuchsia color (first blue well (no growth) for amphotericin B, or first purple well (growth inhibition) or blue well (no growth) for azoles and 5-flucytosine) [27]. The reference strains of *Candida krusei* ATCC^®^ 6258 and *Candida parapsiloisis* ATCC^®^ 22019 were used as quality control strains, as they have defined MIC ranges, according to the M27M44S guideline of the Clinical and Laboratory Standards Institute [28]. The purity of each cell suspension and colony counts were determined by plating 100 µL of the 1.8–9 × 10^3^ cells/mL inoculum on Sabouraud dextrose agar incubated at 35 °C for 48 h.

The ranges of drug concentrations, tested by 2-fold serial dilutions, were 0.125–8 µg/mL for amphotericin B, 0.125–256 µg/mL for fluconazole, 0.015–16 µg/mL for itraconazole, 0.008–8 µg/mL for voriconazole and posaconazole, and 0.06–64 µg/mL for 5-flucytosine. For the whole population, and per molecular type, the frequency of MICs and the geometric mean MICs of each antifungal drug were determined. When available, epidemiologic cutoff values (ECV), defined as the MIC value encompassing at least 97.5% of the wild-type distribution, were used as established elsewhere [29,30]. This was done to determine if the isolates of this study were wild-type or not, to certain antifungal drugs, considering the agreement between commercial test, such as Sensitire^®^ YeastOne^®^ and the CLSI methodology [27,31]. Differences in MICs between VNI and VNII isolates were established, per antifungal drug, using the Mann–Whitney test. All analyses were performed with GraphPad Prism 9 (La Jolla, CA, USA); *p*-values < 0.05 were considered statistically significant.

## 3. Results

### 3.1. The Studied C. neoformans Population from Peru Is Highly Diverse

Among the 26 *C. neoformans* isolates included in this study, 19 (73.1%) were identified as molecular-type VNI and seven (26.9%) as VNII. Among these, eight and three STs were identified, respectively (Table 1).

In VNI, the most common ST was ST32 with six isolates (31.6%), followed by ST5 with four isolates (21.1%), ST69 with three isolates (15.8%), ST1 with two isolates (10.5%) and ST2, ST23, ST31 and ST53 with one isolate each (5.3%). In VNII, the most common ST was ST43 with five isolates (71.4%), followed by ST619 and ST664 with one isolate each (14.3%). Among the loci, 4 different ATs were found in *CAP59*, 4 in *GPD1*, 3 in IGS1, 6 in *LAC1*, 6 in *PLB1*, 3 in *SOD1* and 4 in *URA5* (Table 1). Even though all ATs identified in this study have been reported previously, from the STs, the ST664 is reported here for the first time. The other 10 STs have been reported before in other countries of the world, including ST43, to which the isolate recovered from the 9-year-old girl belongs.

A dendrogram with the MLST sequences, depicting the relationship of the studied isolates with other isolates from Latin America, including 47 clinical, 20 environmental and two isolates of unknow origin from Argentina, Brazil, Colombia and Peru, is shown in Figure 1. Only isolates from these four countries were included, since, until now, MLST studies of *C. neoformans* isolates have not been carried out in other Latin American countries [15]. While most VNI isolates from Peru, grouped with clinical and environmental samples from neighbor countries, VNII isolates, both from Peru and other countries in the region, have not yet been reported on from the environment. In this study, neither VNIII nor VNIV isolates were identified.

Regarding the genetic diversity of the studied isolates, it was possible to determine that, in general, the whole population of *C. neoformans* recovered in three different health care institutions in Lima, Peru, was highly diverse (D = 0.892), although this diversity was mostly due to the VNI population (Table 2). On the other hand, the nucleotide diversity of the Peruvian population was found to be very low (Pi = 0.00667), as only 72 polymorphic sites were found in the alignment of 4044 base pairs (bp) analyzed, with 11 haplotypes or STs that presented a haplotype (gene) diversity, Hd, of 0.892. In Table 2, the values used to assess the genetic diversity of the whole studied population and per molecular type are shown. 

### 3.2. Most of the Studied C. neoformans Isolates from Peru Are Susceptible to Commonly Used Antifungals

The majority of *C. neoformans* clinical isolates from Peru distributed among the wild-type population of the species and the molecular-type VNI, per antifungal drug (Table 3). However, from the 19 VNI isolates, 6 (31.6%) were identified as amphotericin B non-wild-type isolates since they presented an MIC that was one dilution higher than the ECV for this polyene (1 µg/mL). In addition, among these six isolates, two were simultaneously non-wild type for 5-flucytosine, as they also had an MIC that was one dilution higher than the ECV for this antifungal drug (16 µg/mL) (Table 3). For global VNII isolates, the ECVs to the antifungals tested have not be determined, as such, for the Peruvian VNII isolates, it was not possible to establish whether or not these isolates belong to the wild-type distribution of each drug [29,30].

When comparing the geometric mean MICs among molecular types, per antifungal tested, it was possible to establish that the susceptibility of the studied isolates to amphotericin B, 5-flucytosine and posaconazole did not differ depending on whether the isolates were VNI or VNII. However, VNI isolates were found, in a statistically significant way, to be less susceptible to fluconazole (3.214 µg/mL vs. 0.4529 µg/mL, *p* < 0.0001 ****), itraconazole (0.04845 µg/mL vs. 0.02566 µg/mL, *p* = 0.05 *) and voriconazole (0.02911 µg/mL vs. 0.008 µg/mL, *p* < 0.0001 ****) than VNII isolates (Table 3).

## 4. Discussion

The current study shows that in Lima, Peru, as it occurs globally, cryptococcosis affects predominantly immunocompromised male patients, with HIV infection being the main predisposing risk factor and with meningoencephalitis being the main clinical presentation of the infection [1,3]. Our study reports, however, that two cases of cryptococcosis, caused by the molecular-type VNI, occurred in non-HIV infected people, specifically in one patient with cancer and another with a rheumatological disease, which are uncommon risk factors to develop this mycosis [5]. Moreover, this study reports for the first time in Peru, a case of pediatric cryptococcosis, a phenomenon which is even more rarely encountered in the world. In the literature, less than a thousand cases of cryptococcal infection have been reported in children worldwide; in Latin America, no more than 50 cases have been described [32,33,34]. Caused by both *C. neoformans* and *C. gattii* species complexes, cryptococcosis cases in otherwise healthy and HIV-infected children has been documented in Argentina [35], Bolivia [35], Brazil [36,37] and Colombia [33,38]. In the case reported here, a VNII isolate (ST43) was recovered from the 9-year-old Peruvian girl. Although this molecular type has been identified in Colombia as the cause of 8.8% of cases of infection in children (three out of 34 cases), the MLST genotype of these isolates is unknown [33]. Globally, this less frequently seen molecular type, VNII, has been reported, so far, causing pediatric cryptococcosis, in two HIV-infected children from South Africa, an 11-year-old boy and a 10-year-old girl [39]. In these two African cases, specifically, the genotype identified was ST41. This is noteworthy because ST41 is genetically very closely related to ST43, since they differ just by two nucleotides, 1 bp in the *CAP59* gene and 1 bp in the *SOD1* gene.

The genotypic analysis of the Peruvian isolates allowed us as well to identify that the molecular-type VNI, as reported previously in Peru, in Latin America and globally [1,10,34], is the most common molecular type causing cryptococcosis. However, our study differs with respect to the degree of genetic diversity found previously among the *C. neoformans* population. In Peru, low genetic diversity (D = 0.17) was revealed in a MLST analysis of 48 isolates from 20 patients. This was most likely because that particular study was carried out with at least two consecutive isolates recovered from the same patient, leading to the identification of the same genotype in serial isolates, as occurred in 70% of cases [10]. In Latin America, the genetic diversity of *C. neoformans* isolates, which was lower than the one calculated for *C. gattii* isolates (D = 0.7104 vs. 0.9755), was also lower than the diversity reported in our study (D = 0.892) [15]. In general, and according to reports from the ISHAM Working Group “Genotyping of *Cryptococcus neoformans* and *C. gattii*”, *C. neoformans* spreads rather clonally. This is since, among the more than 600 STs identified until now in this species complex, only seven VNI STs (ST5, ST93, ST4, ST23, ST31, ST6 and ST32, in order of frequency) account for almost 60% of all *C. neoformans* isolates in the world [40].

Among the STs identified in this study, ST32, which was the most frequent one, has not only been identified in other clinical isolates in Peru [10], and the region, in countries like Brazil [18,20] and Colombia [23], but it has also been reported in human cases in the USA [18], Europe, in France [41] and Germany [42], Asia, in China [43], Japan [44,45], Thailand [46,47] and Vietnam [48], and Africa, in Botswana [49], South Africa [39], Tanzania, Uganda and Zaire [18]. ST32 has also been recovered from environmental samples in Belgium [18] and from veterinary cases of cryptococcosis, in one cat and two dogs in the USA [50]. ST5, which is the most common ST reported worldwide, accounting alone for more than 30% of clinical, environmental and veterinary isolates [40], has been reported as causing human cryptococcosis in the countries already mentioned for ST32, except Zaire and Tanzania, plus Greece [51], Italy [18,52], and Spain [51] in Europe, India [53], Korea [54], Kuwait [46], Laos [55], Qatar [46], and South Korea [56] in Asia, and Ivory Coast [57] and Malawi [18] in Africa. This ST5 was also responsible for one case of feline cryptococcosis in the USA [50] and another one in Italy [52]. The third most common ST found in this study, ST69, is among the 10 more common STs found globally. Although ST69 has been most frequently reported from South Africa [58], clinical cases caused by this genotype have also occurred in Colombia [23], China [59], France [41], Germany [42], Indonesia and Kuwait [46], Italy [60] and Uganda [61], as well as in a unique report from the environment in Greece [51]. ST1 has remarkably been reported from clinical samples only in Peru [10]. However, in France and the USA, ST1 has been recovered from pigeon droppings and other environmental sources [18]. From the STs that were identified in one isolate each in our study, ST2, ST23 and ST31 are also among the 10 more common STs found globally in clinical, environmental and veterinary isolates [40]. Conversely, ST53 has a more restricted distribution, with reports made with clinical and environmental isolates from Brazil [19], China [46,62] and Thailand [46,63]. From the five most frequent STs reported in *C. neoformans* isolates from Latin America (ST93, ST77, ST2, ST5 and ST23) [15], ST93 and ST77 were not identified in Peru.

Among the STs identified in VNII isolates from Peru, the most common was ST43, which has not been reported in neighboring countries. However, it has been identified in human samples from Germany [42], Italy [64], Japan [44], Nigeria [64,65] South Africa [39,58], Thailand [63,66] and the USA [18], from the environment in Nigeria [64,65] and the USA [63] and from cats in the USA and Thailand [66]. ST619, on the other hand, was just recently reported in a single veterinary case in Brazil, isolated from the skin lesion of a domestic cat [67]. Although ST664 was identified for the first time in this study, this ST is genetically very closely related to ST41, an ST more common among VNII isolates, since they differ just by three nucleotides in the *PLB1* gen. 

The identification of six VNI isolates that are not wild-type for amphotericin B in this study is very important as, in Peru, the induction therapy for cryptococcosis has been reported to consist of amphotericin B deoxycholate alone or in combination with fluconazole in the majority of cases [68,69]. Even though acquired resistance to polyenes is uncommon in *C. neoformans*, a reduced susceptibility to drugs such as amphotericin B, after its use to treat cryptococcal meningitis, has been reported for more than two decades [70]. In addition, amphotericin B non-wild-type isolates were recently identified in five *C. gattii* VGI isolates from Australia, the Democratic Republic of Congo, Malaysia, and interestingly, Peru [71]. The identification of two VNI isolates that are in addition non-wildtype for 5-flucytosine draws attention. In the few cases worldwide where high MICs to 5-flucytosine have been reported, it has been in patients with prolonged exposure to the antifungal medication or with monotherapy [12,72]. This is not the case of Peru, as this antifungal drug is still unavailable in the country [69]. Reduced susceptibility of the Peruvian isolates to 5-flucytosine could perhaps be due to unknown mechanisms that need further exploration. Nevertheless, considering that ECVs do not always predict a therapeutic response and that, for both amphotericin B and 5-flucytosine, non-wildtype isolates isolates were close to the borderline of the ECVs, it is possible that these six isolates are still treatable with conventional therapies.

Regarding the variation in the antifungal susceptibility profile of the studied isolates, depending on the molecular type, it was possible to determine that the Peruvian VNI isolates were less susceptible to three azoles—fluconazole, itraconazole and voriconazole—than VNII isolates. While differential antifungal susceptibility to fluconazole and other azoles has been widely reported in the molecular types of both *C. neoformans* and *C. gattii* [73,74,75,76], comparisons between VNI and VNII isolates have been rarely performed. Most studies carrying out susceptibility testing do not identify the molecular type of the isolates. As such, comparisons cannot be performed [29]. In addition, VNII isolates are less commonly recovered worldwide [77], and therefore susceptibility data on these isolates is scarce. In Latin America, for instance, VNII accounted for less than 3% of isolates with data on antifungal susceptibility [15]. Until now, one study from Australia and one from Denmark described a statistically significant difference between VNI and VNII isolates tested against fluconazole, with a higher geometric mean MIC in the VNI group [78,79], which is in agreement with our findings. However, in both studies, no other differences with statistical significance were found among these two molecular types, even though VNI tended to be less susceptible to most of the antifungals tested.

To conclude, this study contributes with data on the molecular epidemiology of the main etiological agent of cryptococcosis, *C. neoformans*, not only with respect to the major molecular type VNI, but also, to the less frequently recovered VNII. Unlike VNI isolates, VNII are generally rarely recovered from the environment, with only eight isolates reported so far, from Libya, Nigeria and the USA [51,63,64,65]. As such, our study encourages researchers to carry out environmental studies in Peru, to recognize the saprophytic source of VNII isolates causing cryptococcosis in the country. Importantly, this study documents the first case of pediatric cryptococcal infection in Peru, with data on the genotype and antifungal susceptibility of the isolate recovered. The determination of the susceptibility profiles of the studied isolates, highlighting the identification of non-wild type isolates for amphotericin B and 5-flucytosine, also emphasizes the need of surveilling the emergence of isolates with reduced susceptibility to commonly used antifungal drugs, as this can lead to treatment failure. Studies, like the one reported herein on the epidemiology of cryptococcosis and the genotypes and antifungal susceptibility profiles of *C. neoformans* isolates, are very important, since this mycosis remains an opportunistic infection with a significant burden in Peru, where there is a constant increase in HIV seroprevalence and many areas of the country are resource-limited. 

## Figures and Tables

**Figure 1 jof-08-01306-f001:**
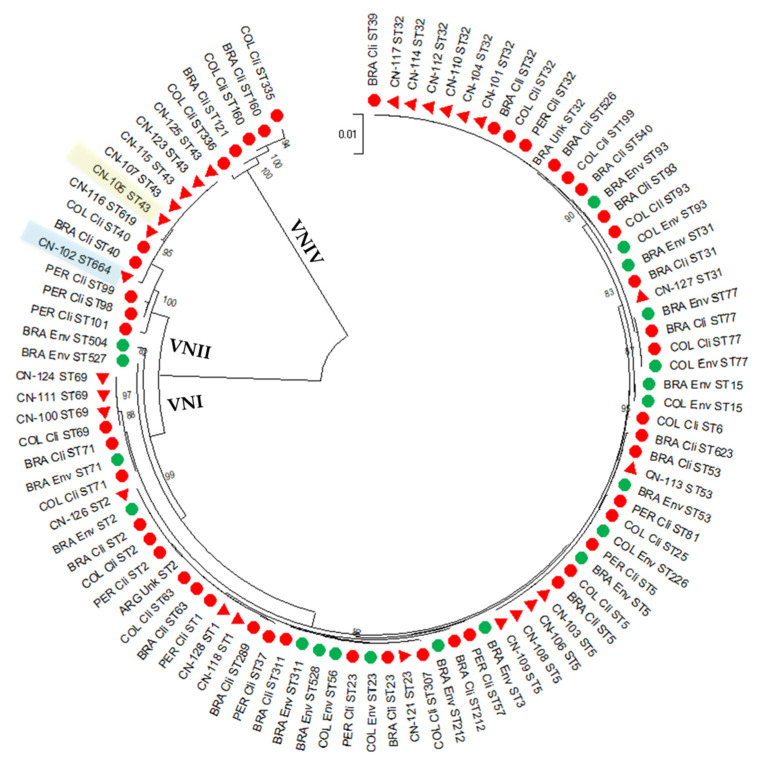
A dendrogram depicting the genetic relationship between clinical *Cryptococcus neoformans* isolates from this study (triangles) and other clinical (red) and environmental (green) isolates from Argentina (ARG), Brazil (BRA), Colombia (COL) and Peru (PER), as reported in former *C. neoformans* genotyping studies [10,18,19,20,21,22,23]. The VNII sequence types (ST), ST664, newly identified, and ST43, belonging to an isolate from a 9-year-old girl, are highlighted in blue and yellow, respectively. The dendrogram is based on the analysis of the seven concatenated ISHAM consensus MLST loci using the program MEGA 11 [24]. Bootstrap values above 75 are indicated on the branches.

**Table 1 jof-08-01306-t001:** Epidemiological and genotypic data of clinical *Cryptococcus neoformans* isolates from Peru.

Isolate	Year	Age (Years) ^1^	Sex ^1,2^	*CAP59*	*GPD1*	IGS1	*LAC1*	*PLB1*	*SOD1*	*URA5*	ST ^3^	MT ^4^
CN-100	2012	NA	NA	7	5	1	3	3	1	1	69	VNI
CN-101	2013	NA	NA	1	1	10	3	4	1	1	32	VNI
CN-102	2013	NA	NA	10	9	14	8	12	12	4	664 ^5^	VNII
CN-103	2014	43	M	1	3	1	5	2	1	1	5	VNI
CN-104	2014	47	M	1	1	10	3	4	1	1	32	VNI
**CN-105**	**2014**	**9**	**F**	**2**	**9**	**14**	**8**	**11**	**11**	**4**	**43**	**VNII**
CN-106	2015	39	M	1	3	1	5	2	1	1	5	VNI
CN-107	2017	NA	NA	2	9	14	8	11	11	4	43	VNII
CN-108	2017	NA	NA	1	3	1	5	2	1	1	5	VNI
CN-109	2018	36	M	1	3	1	5	2	1	1	5	VNI
CN-110	2018	41	M	1	1	10	3	4	1	1	32	VNI
CN-111	2018	36	M	7	5	1	3	3	1	1	69	VNI
CN-112	2019	45	M	1	1	10	3	4	1	1	32	VNI
CN-113	2019	54	M	1	3	1	3	2	1	1	53	VNI
CN-114 ^6^	2019	71	F	1	1	10	3	4	1	1	32	VNI
CN-115	2019	35	M	2	9	14	8	11	11	4	43	VNII
CN-116	2019	63	M	2	9	14	11	11	11	15	619	VNII
CN-117	2019	31	M	1	1	10	3	4	1	1	32	VNI
CN-118 ^7^	2019	70	M	7	1	1	1	1	1	1	1	VNI
CN-121	2020	35	M	7	1	1	2	1	1	2	23	VNI
CN-123	2020	29	M	2	9	14	8	11	11	4	43	VNII
CN-124	2020	31	M	7	5	1	3	3	1	1	69	VNI
CN-125	2020	25	M	2	9	14	8	11	11	4	43	VNII
CN-126	2020	51	M	7	1	1	1	1	1	2	2	VNI
CN-127	2020	59	M	1	1	10	3	2	1	1	31	VNI
CN-128	2021	45	M	7	1	1	1	1	1	1	1	VNI

^1^ NA: not available; ^2^ F: female, M: male, ^3^ ST: sequence type; ^4^ MT: molecular type; ^5^ newly identified ST; ^6^ patient with cancer; ^7^ patient with a rheumatologic disease. The pediatric case is in bold.

**Table 2 jof-08-01306-t002:** Genotypic diversity of clinical *Cryptococcus neoformans* isolates from Peru, showing the values for the whole population and per molecular type.

Population	n	D ^1^	SS ^2^	H ^3^	Hd ^4^	Pi ^5^
All	26	0.892	72 bp	11	0.892	0.00667
VNI	19	0.854	24 bp	8	0.854	0.00233
VNII	7	0.524	9 bp	3	0.524	0.00064

^1^ D: Simpson’s diversity index; ^2^ SS: Number of polymorphic (segregating) sites; ^3^ h: number of haplotypes (sequence types); ^4^ Hd: haplotype (gene) diversity; ^5^ Pi: nucleotide diversity.

**Table 3 jof-08-01306-t003:** Distribution of the minimal inhibitory concentrations (MIC) of clinical *Cryptococcus neoformans* isolates from Peru. Modes are underlined. Non-wild-type isolates, which were determined with results obtained with the Clinical and Laboratory Standards Institute method, as referenced elsewhere [29,30], are highlighted.

				No. of Isolates at MIC Value (µg/mL)											
Antifungal	MT ^1^	n	GM ^2^ (µg/mL)	0.008	0.015	0.03	0.06	0.12	0.25	0.5	1	2	4	8	16
Amphotericin B	VNI	19	0.4821					1	5	7	6				
	VNII	7	0.5000						3	2	1	1			
	All	26	0.4868					1	8	9	7	1			
5-flucytosine	VNI	19	2.9880							2	1	6	6	2	2
	VNII	7	1.0000				1			2		2	2		
	All	26	2.2250				1			4	1	8	8	2	2
Fluconazole ^3^	VNI	19	3.2140								1	7	8	3	
	VNII	7	0.4529					1		5	1				
	All	26	1.8960					1		5	2	7	8	3	
Itraconazole ^3^	VNI	19	0.0485		2	8	4	5							
	VNII	7	0.0257		3	3	1								
	All	26	0.0408		5	11	5	5							
Voriconazole ^3^	VNI	19	0.0291	2	4	7	6								
	VNII	7	0.0080	7											
	All	26	0.0206	9	4	7	6								
Posaconazole	VNI	19	0.0629	1		5	6	6	1						
	VNII	7	0.0381		1	4	1	1							
	All	26	0.0548	1	1	9	7	7	1						

^1^ MT: molecular type; ^2^ GM: geometric mean; ^3^ statistically significant differences between molecular types (*p* < 0.05) were found.

## Data Availability

The individual data of the isolates included in this study are available from the corresponding author upon request. The sequences obtained in this study were deposited in GenBank.

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
