# Peer review of "Cryptococcus neoformans Causing Meningoencephalitis in Adults and a Child from Lima, Peru: Genotypic Diversity and Antifungal Susceptibility"

_jof, 2022, doi:10.3390/jof8121306_

Round 1
Reviewer 1 Report
The manuscript is well written and interesting. It provides more information on the distribution of yeasts of the C. neoformans species complex. This type of article is very important and necessary when working on the molecular epidemiology of these yeasts.
Line 66 :
Despite the currently available options for the treatment of cryptococcosis, resistance to some antifungals has been described in isolates of both C. neoformans and C. gattii species complexes worldwide [11].
Please add a reference for C. gattii resistance to antifungals
Line 129
Why the sensitivity test was made using the Sensi-129 titre® YeastOne® plates and not the CLSI method which is the reference technique for fungi.
If the authors use a commercial method, justify by references that other studies have already published by comparing the MIC obtained with the commercial method Sensititre to the ECV obtained with the CLSI method.
Did the authors study by MLST one colony from the culture of patients' CSF on Sabouraud medium, or several colonies from the sample. We know that in the same sample we can have mixtures of strains of variable ST but also with different susceptibility to antifungals.
In table 3: In my opinion, it is not necessary to give the repartition of MIC for all strains "all" for each antifungal because it is better to have the repartition of MIC by genotype.
Line 317 :
“While differential antifungal susceptibility to fluconazole and other 317 azoles has been widely reported in the molecular types of both C. neoformans and C. gattii 318 [69-72], comparisons between VNI and VNII isolates have been rarely done.”
You must add reference 54 again because Kassi et al in their study in Ivory Coast have found VNII and VNI strains, they give the ST and they mentioned the susceptibility of the isolates to AMB, 5FC and fluconazole.
About the susceptibility to AMB, the non-wild type profile is uncommon but finally more and more described in the literature and also with strains from the C. gattii complex species (bellet et al., 2022).
About the susceptibility to 5FC, non wild type was found even if this antifungal is not used in some countriesAuthor Response
Comments and Suggestions for Authors
The manuscript is well written and interesting. It provides more information on the distribution of yeasts of the C. neoformans species complex. This type of article is very important and necessary when working on the molecular epidemiology of these yeasts.
Answer: thanks to the reviewer for the time to read and evaluate the manuscript and for each of the valuable comments and suggestions. In the revised manuscript and along with this point-by-point response, we aimed to address all issues mentioned by the reviewer.
Line 66 :
Despite the currently available options for the treatment of cryptococcosis, resistance to some antifungals has been described in isolates of both C. neoformans and C. gattii species complexes worldwide [11].
Please add a reference for C. gattii resistance to antifungals.
Answer: following the suggestion of the reviewer, two additional references were added (Billmyre et al 2017, and Carneiro et al. 2020).
Line 129
Why the sensitivity test was made using the Sensi-129 titre® YeastOne® plates and not the CLSI method which is the reference technique for fungi.
If the authors use a commercial method, justify by references that other studies have already published by comparing the MIC obtained with the commercial method Sensititre to the ECV obtained with the CLSI method.
Answer: thanks to the reviewer for this question. Unfortunately, in many Latin-American countries, including Peru and Colombia, where the experiments were done, it is very difficult and expensive to purchase some antifungal drugs, like 5-flucytosine and posaconazole, to carry out susceptibility testing as it is recommended by the CLSI. In this way, the use of a commercial test, such as YeastOne, offers us the possibility to test 9 different antifungals (6 in the case of Cryptococcus) at the same time. In addition, as it is already mentioned in the manuscript, a study done by well-known researchers working with susceptibility testing for fungal pathogens (Espinel-Ingroff et al. J Clin Microbiol 1999, 37, 591-595), helps us to justify the use of the YeastOne panel, as this study suggested the potential value of this commercial test even for use in the clinical laboratory. Nevertheless, to further justify the use of a commercial method, a new reference was added as suggested by the reviewer (Delma et al. 2020).
Did the authors study by MLST one colony from the culture of patients' CSF on Sabouraud medium, or several colonies from the sample. We know that in the same sample we can have mixtures of strains of variable ST but also with different susceptibility to antifungals.
Answer: we agree with the reviewer. We have seen that different colonies from the same clinical sample can belong to a different genotype and have different susceptibility profile. As such, our study was done with single colonies. This is now mentioned in the text.
In table 3: In my opinion, it is not necessary to give the repartition of MIC for all strains "all" for each antifungal because it is better to have the repartition of MIC by genotype.
Answer: we decided to include “all” to be able to show a general idea on the susceptibility of the species complex as it is (C. neoformans), because many times the molecular type is not determined and data on antifungal susceptibility is given to C. neoformans “without genotype”. In fact, the determination of EVCs in this species complex is based, in a big part, on nontyped isolates (according to 2 publications from Espinel-Ingroff 2012). In this way, we considered that these values should stay in Table 3.
Line 317 :
“While differential antifungal susceptibility to fluconazole and other azoles has been widely reported in the molecular types of both C. neoformans and C. gattii [69-72], comparisons between VNI and VNII isolates have been rarely done.”
You must add reference 54 again because Kassi et al in their study in Ivory Coast have found VNII and VNI strains, they give the ST and they mentioned the susceptibility of the isolates to AMB, 5FC and fluconazole.
Answer: while Kassi et al. performed susceptibility testing for VNI and VNII strains, they did not carry out comparisons between the MIC of each molecular type. As such, in Line 317, there is no need to mention this study. The authors conclude that “no correlation between VNI and VNII STs and an elevated MIC to FCZ was found”, but they do not conclude about VNI being more, or less susceptible to the tested antifungals, than VNII.
About the susceptibility to AMB, the non-wild type profile is uncommon but finally more and more described in the literature and also with strains from the C. gattii complex species (bellet et al., 2022). About the susceptibility to 5FC, non wild type was found even if this antifungal is not used in some countries.
Answer: following the suggestion of the reviewer, the new reference was added, to report these interesting findings.
Reviewer 2 Report
Comments to authors
Firacative et al. present the molecular typing (MLST) and susceptibility profile of 26 C. neoformans isolated in Peru (Lima) from 2012 to 2021.
They found 73% of VNI relative to VNII (27%). This is similar to what reported for Peru in a review of the Latin American Cryptococcus isolates (ref 13) published in 2021 with a ratio of 75% to 25% for VNI to VNII. A similar study was also published in 2015 (ref 9) with isolates from Lima, but genotyping performed by AFLP method. Another one in 2020 (ref 10)
A few isolates found to be non-wiltype, but most MIC borderline with ECV.
The paper well written and the analyses sound.
Major/General Comments:
1. Can the authors clarify if the isolates analyzed are the same as the one reported in Bejar et al, 2015 or Van de Wiele et al published in 2020? If so, what proportion of the 26 isolates of the current study? I think it would be important to disclose in the manuscript, to confirm the that MLST data is novel. C. Firacative is co-author on the 2020 study.
2. Was any of the patients immunocompetent or without comorbidity that would put them at risk of cryptococcal infection?
3. Figure 1 is rather difficult to read with red triangles and circles to differentiate isolates of the study vs. the reference ones. A different colour could be used. I would also suggest to colour code the countries in an extra exterior layer for a better visual summary.
4. It is not clear to me which ECV were used. It may be best to summarize in a supplemental table and reference. CLSI ECVs are found in M57S (please reference), but were not assessed with Yeast One methodology (which could create a bias). A note to that limitation should be included.
5. Yeast One panels must be read at first colour change except for amphotericin (no colour change). At line 139, the authors indicate the plates were read at «lowest MIC that prevented colour change» which I interpret that they incorrectly read first well with no colour change. Please clarify or correct.
Minor/Specific Comments:
1. Abstract ln 18. Suggestion: Indicate «Yeast One broth microdilution» for added precision.
2. Intro ln 70: correct «might hindrance» for «might hinder».
3. Mat and meth, ln 125: The NCBI sequences are not currently available on Genbank. Please ensure they are upon resubmission.
4. Mat and meth, ln 150: The currently accepted CLSI ECV cutoff is 97.5%. I would advise using that cutoff. Please also reference the latest release of M57S ED4 document in which CLSI ECVs listed for C. neoformans VNI.
5. Results, ln 157: correct «higly» for «highly»
6. Results, table 1: Was any trend seen in molecular types or susceptibility data according to the year of sampling?
7. Ln 218-220, Table 3 and elsewhere: remove commas for decimal and use a dot.
8. Results, ln 205-214: Indicate that NWT MICs all borderline (1 dil) from ECV. At those MIC, the isolates might likely be treatable.
9. Discussion, ln 234: change «has been informed» for has «has been documented»
10. Table 3: clarify what ECV cutoffs were specifically used and from what source (add ref to M57S ED4 2022)
11. Discussion, Ln265-292. It would be nice to summarize this in an extra table with prevalence of those discussed ST in Peru, vs. Latin America and Worlwide.
12. Discussion Ln 308-312: As stated above, the MICs are borderline with ECV. Might still be treatable. Add that nuance.
Author Response
Comments to authors
Firacative et al. present the molecular typing (MLST) and susceptibility profile of 26 C. neoformans isolated in Peru (Lima) from 2012 to 2021.
They found 73% of VNI relative to VNII (27%). This is similar to what reported for Peru in a review of the Latin American Cryptococcus isolates (ref 13) published in 2021 with a ratio of 75% to 25% for VNI to VNII. A similar study was also published in 2015 (ref 9) with isolates from Lima, but genotyping performed by AFLP method. Another one in 2020 (ref 10)
A few isolates found to be non-wiltype, but most MIC borderline with ECV.
The paper well written and the analyses sound.
Answer: thanks to the reviewer for the time to read and evaluate the manuscript and for each of the valuable comments and suggestions. In the revised manuscript and along with this point-by-point response, we aimed to address all issues mentioned by the reviewer.
Major/General Comments:
- Can the authors clarify if the isolates analyzed are the same as the one reported in Bejar et al, 2015 or Van de Wiele et al published in 2020? If so, what proportion of the 26 isolates of the current study? I think it would be important to disclose in the manuscript, to confirm the that MLST data is novel. C. Firacative is co-author on the 2020 study.
Answer: thanks to the reviewer for this question. None of the isolates reported in this manuscript had been studied before. The study from Bejar et al, 2015, reports isolates recovered between Jan 2018 and Dec 2012. In the study by van de Wiele, isolates from Jun 1997 to Jun 2002 were included. Our isolates were recovered between 2012 (only 1 isolate from this year) and 2021. As such, we consider, that there is no need to mention in the manuscript that the MLST data is novel.
- Was any of the patients immunocompetent or without comorbidity that would put them at risk of cryptococcal infection?
Answer: No. All patients were considered immunocompromised as they had HIV (92.3%), cancer (3.8%) or a rheumatologic disease (3.8%). This is mentioned in the text (Line 92) as well as in Table 1.
- Figure 1 is rather difficult to read with red triangles and circles to differentiate isolates of the study vs. the reference ones. A different colour could be used. I would also suggest to colour code the countries in an extra exterior layer for a better visual summary.
Answer: thanks to the reviewer for this suggestion. The colour, red and green, is used to indicate clinical and environmental isolates, respectively, in order to emphasize the relatedness among these two sources. Triangles were the best way to indicate isolates from this study (compared to squares and diamonds). Unfortunately, the software we use to generate the dendrogram does not allow to change the colour of the font and we do not have any other program to edit this, as such, we used the 3-letter code. In this way, we consider we can keep the figure as it is.
- It is not clear to me which ECV were used. It may be best to summarize in a supplemental table and reference. CLSI ECVs are found in M57S (please reference), but were not assessed with Yeast One methodology (which could create a bias). A note to that limitation should be included.
Answer: ECVs for C. neoformans reported in M57S come from two citations mentioned in this manuscript (both from Espinel-Ingroff 2012). With respect to the use of Yeast One, as it was answered as well to the other reviewer, we would like to refer to a study done by well-known researchers working with susceptibility testing for fungal pathogens (Espinel-Ingroff et al. J Clin Microbiol 1999, 37, 591-595). This study helps us to justify the use of this commercial test, as it suggests the potential value of this panel even for use in the clinical laboratory. In addition, to further justify the use of a commercial method, a new reference was added as suggested by the reviewer 1 (Delma et al. 2020). As only 2 types of non-wild type isolates were identified (for AMB and 5-FC), we considered that it is not necessary to create another table, as, in the legend of table 3, the definition of non-wild type isolates is indicated.
- Yeast One panels must be read at first colour change except for amphotericin (no colour change). At line 139, the authors indicate the plates were read at «lowest MIC that prevented colour change» which I interpret that they incorrectly read first well with no colour change. Please clarify or correct.
Answer: thanks to the reviewer for noticing this. When reading the plates, we did consider the difference between amphotericin B vs. azoles and 5-FC. To clarify this, we verified the way it is written in Espinel-Ingroff et al. J Clin Microbiol 1999, 37, 591-595, and made the changes in the manuscript accordingly.
Minor/Specific Comments:
- Abstract ln 18. Suggestion: Indicate «Yeast One broth microdilution» for added precision.
- Intro ln 70: correct «might hindrance» for «might hinder».
Answer: the changes were done following the suggestions of the reviewer.
- Mat and meth, ln 125: The NCBI sequences are not currently available on Genbank. Please ensure they are upon resubmission.
Answer: submitted sequences will not be publicly available, at least the publication is accepted. This is a suggestion from GenBank: “You may prepare and submit your manuscript before your accessions are released in GenBank. They (accession numbers) will not be released to the public database until the data or accession numbers appear in print”. Once our manuscript is accepted, we will make sure that accesion numbers are available.
- Mat and meth, ln 150: The currently accepted CLSI ECV cutoff is 97.5%. I would advise using that cutoff. Please also reference the latest release of M57S ED4 document in which CLSI ECVs listed for C. neoformans VNI.
Answer: thanks to the reviewer for this suggestion. ECV of 97.5% is used now. As we mentioned in a previous comment, please consider that the ECVs for C. neoformans reported in M57S come from two citations mentioned in this manuscript (both from Espinel-Ingroff 2012).
- Results, ln 157: correct «higly» for «highly»
Answer: the change was done following the suggestion of the reviewer.
- Results, table 1: Was any trend seen in molecular types or susceptibility data according to the year of sampling?
Answer: Unfortunately, they were not enough isolates per year to carry out these comparisons. As such, it was not possible to calculate any trend.
- Ln 218-220, Table 3 and elsewhere: remove commas for decimal and use a dot.
Answer: Thanks to the reviewer for noticing this. The changes were done accordingly.
- Results, ln 205-214: Indicate that NWT MICs all borderline (1 dil) from ECV. At those MIC, the isolates might likely be treatable.
Answer: the change was done, in the discussion section, following this and the last (#12) suggestion of the reviewer.
- Discussion, ln 234: change «has been informed» for has «has been documented»
Answer: the change was done following the suggestion of the reviewer.
- Table 3: clarify what ECV cutoffs were specifically used and from what source (add ref to M57S ED4 2022)
Answer: ECVs for C. neoformans reported in M57S come from two citations mentioned in this manuscript (both from Espinel-Ingroff 2012). As only 2 types of non-wild type isolates were identified (for AMB and 5-FC), we considered that it is not necessary to add additional information in table 3, as in its legend, the definition of non-wild type isolates is indicated.
- Discussion, Ln265-292. It would be nice to summarize this in an extra table with prevalence of those discussed ST in Peru, vs. Latin America and Worlwide.
Answer: while we agree with the reviewer about including the prevalence of STs in Peru and Latin America and the world, this analysis will be difficult to do. Many times there is no information on the strain IDs per country and some strains might be included in more than one study, for instance, when they are first reported in a local study (a country) and then in global population studies. As such, giving numbers with the prevalence of certain STs will not be reliable. We consider that, so far, we can just mention if an ST has been reported elsewhere, but not in which proportion.
- Discussion Ln 308-312: As stated above, the MICs are borderline with ECV. Might still be treatable. Add that nuance.
Answer: the change was done following this and the suggestion #8 of the reviewer.